# Improving Anti-Coking Properties of Ni/Al_2_O_3_ Catalysts via Synergistic Effect of Metallic Nickel and Nickel Phosphides in Dry Methane Reforming

**DOI:** 10.3390/ma15093044

**Published:** 2022-04-22

**Authors:** Yu Shi, Shiwei Wang, Yiming Li, Fan Yang, Hongbo Yu, Yuting Chu, Tong Li, Hongfeng Yin

**Affiliations:** Ningbo Institute of Materials Technology and Engineering, Chinese Academy of Sciences, 1219 Zhongguan West Road, Ningbo 315201, China; shiyu@nimte.ac.cn (Y.S.); wangshiwei@nimte.ac.cn (S.W.); liyiming@nimte.ac.cn (Y.L.); yf@minte.ac.cn (F.Y.); yuhongbo@nimte.ac.cn (H.Y.); chuyuting@nimte.ac.cn (Y.C.)

**Keywords:** dry methane reforming, nickel catalysts, phosphides, coke-resistance, synergistic effect

## Abstract

A series of NiP-x/Al_2_O_3_ catalysts containing different ratio of metallic nickel to nickel phosphides, prepared by varying Ni/P molar ratio of 4, 3, 2 through a co-impregnation method, were employed to investigate the synergistic effect of metallic nickel-nickel phosphides in dry methane reforming reaction. The Ni/Al_2_O_3_ catalyst indicates good activity along with severe carbon deposition. The presence of phosphorus increases nickel dispersion as well as the interaction between nickel and alumina support, which results in smaller nickel particles. The co-existence of metallic nickel and nickel phosphides species is confirmed at all the P contained catalysts. Due to the relative stronger CO_2_ dissociation ability, the NiP-x/Al_2_O_3_ catalysts indicate obvious higher resistance of carbon deposition. Furthermore, because of good balance between CH_4_ dissociation and CO_2_ dissociation, NiP-2/Al_2_O_3_ catalyst exhibits best resistance of carbon deposition, few carbon depositions were formed after 50 h of dry methane reforming.

## 1. Introduction

Dry reforming of methane, described as CH_4_ + CO_2_ = 2H_2_ + 2CO, has been intensively studied since it was reported [1]. It has two obvious advantages: the reactants of dry methane reforming are two kinds of greenhouse gases (CH_4_ and CO_2_). On the other hand, the low H_2_/CO ratio for the products of dry methane reforming (DMR) is suitable for the Fischer–Tropsch synthesis and methanol synthesis [2,3].

Catalysts play crucial roles during dry methane reforming process. Noble metals, such as Ru, Rh, Pd, Pt, and Ir, display excellent catalytic performance and good stability in dry methane reforming, whereas the limits of resources and high costs restrict their utilization scales on the industry. Transition metals, such as Fe, Cu, Co, especially Ni, have been proved a kind of efficient catalysts for dry methane reforming [4,5]. However, due to the relative stronger methane dissociation ability of transition metals, the carbon deposition over transition metals is a crucial problem which is the restriction for the utilization [6].

As is shown in the literatures [7], carbon deposition on the surface of nickel catalyst is generally carried out according to the following steps: methane molecules dissociate on the metal surface to generate adsorbed CH_x−_* species, which will be further dehydrogenated to form C_−_*; part of the C_−_* species can be removed by the surface active O_−_* species generated from the dissociation of CO_2_ molecules and convert into CO; at the same time, the unconverted CH_x−_* will further undergo deep cracking on the metal surface to form surface carbon; In addition, the further aggregation of adsorbed CH_x−_* species is another way to form carbon deposits. It follows that increasing the adsorption and dissociation rate of CO_2_ and appropriately reducing the methane dissociation ability of the catalyst are effective ways to inhibit the carbon deposition on the surface of nickel catalyst.

Many endeavors have been studied to suppress the carbon deposition over nickel catalyst in dry methane reforming. These studies mainly focus on: (1) Enhance the surface basicity of the catalysts by the addition of alkaline promoters, which increases the CO_2_ dissociation ability, promotes the generation of surface-active O_−_* species and enhances the elimination of surface C_−_* species on nickel catalysts. The promotion of alkalis [8], alkaline-earth metals [9,10], and rare-earth metals (La, Ce) [11,12] can enhance the dispersion of nickel species and the surface basicity and promote the generation of surface active O_−_* species from CO_2_, which results in better carbon deposition resistance over nickel catalysts, whereas the enhancement of surface basicity always accompanies the decrease of methane dissociation rate, which leads to the decrease of activity [8]. (2) Obtain smaller nickel nanoparticle size by confining nickel species with mesoporous materials [13,14], inorganic shells [11,15], and the edge of the support [3], which is beneficial of the resistance for carbon formation during DMR. Wang et al. [13] investigated S-2 zeolite fixed nickel nanoparticles structure catalyst (Ni@S-2) in DMR and found Ni@S-2 exhibited superb carbon resistance. Liu et al. [16] synthesized a multiple-core@shell structured catalyst (Ni-ZrO_2_@SiO_2_) catalyst and investigated its catalytic performance in DMR. Due to the small nickel nanoparticle size and the confinement effect of SiO_2_ shell, Ni-ZrO_2_@SiO_2_ displayed ultra-high coking resistance for dry methane reforming. Song et al. [3] found that the edges of MgO crystals exhibited good properties of the stabilization of nickel nanoparticles, which resulted in good resistance of carbon deposition in DMR. What is more, special structure catalysts, such as spinels (NiAl_2_O_4_ [17], Ni/MgAl_2_O_4_ [18]), perovskites (LaNiO_3_ [19], La_1−x_Pr_x_NiO_3−δ_ [20], La_0.6_Sr_0.2_Ti_0.85_Ni_0.15_O_3−δ_ [21]), hydrotalcites (Ce-NiMgAl hydrotalcite [22], Zr-NiMgAl hydrotalcite [23], La-NiMgAl hydrotalcite [24]), and solid solutions (NiOMgO [25], NiO-CeO_2_ [26]), which have a strong ability for confining nickel nanoparticles, are also used to increase the carbon resistance of nickel catalysts; (3) Decrease the outer electron cloud density of nickel catalyst by alloying nickel with other metals [27,28], which decreases CH_4_ dissociation ability of the catalysts. Turap et al. [27] indicated that Co-Ni alloy promoted the adsorption of surface oxygen and enhanced carbon removal, resulting in less carbon deposition compared to nickel catalyst; (4) Poison CH_4_ dissociation ability of nickel with sulfur [29], which, however, passivates the activity of nickel catalysts in dry methane reforming.

Transition metal carbides, such as Mo_2_C and WC, which have similar outer electronic structure to noble metals, are becoming an attractive and promising catalyst for replacement of traditional dry methane reforming catalysts due to their excellent resistance to carbon deposition [30,31,32]. However, due to the relative strong CO_2_ dissociation properties, transition metal carbides are easy oxidized by CO_2_ to inactive oxides phase, which leads to rapid deactivation during dry methane reforming process at atmospheric pressure. Subsequently, Ni–Mo_2_C catalysts were found to show stable DMR activity at atmospheric pressure [32,33]. Ni species are proposed to be in charge of the dissociation of CH_4_, while Mo_2_C is responsible for the dissociation of CO_2_. However, the appropriate molar ratio of Ni to carbides is controversial in different literatures. Shi et al. [33] reported that the optimum Ni/Mo_2_C ratio was 1/2, while Cheng et al. [34] reported that the catalyst with Ni/Mo_2_C = 1/10 exhibited best stability.

Recently, transition metal phosphides (MoP, WP, NiP_x_) were demonstrated to exhibit better activity, stability, and higher resistance of oxidation and coking than Ni/Mo_2_C catalyst in dry methane reforming [35,36]. However, transition metals phosphide catalysts still exhibit relative stronger CO_2_ dissociation properties than CH_4_ dissociation properties. Yao et al. [35] found that bulk MoP catalyst preferred low temperature and space velocity for DMR, the oxidation by CO_2_ to MoO_2_ species was also observed at the condition of relative high temperature and WHSV. González-Castaño et al. [37] investigated the catalytic performance of 20 wt% Ni_2_P supported on Al_2_O_3_, CeO_2_, and SiO_2_-Al_2_O_3_, and the result indicated that the Ni_2_P/Al_2_O_3_ catalyst showed highest conversion and stability. However, visible deactivation was still observed after 10 h of time on stream, which was due to the oxidation of Ni_2_P phase during dry methane reforming. Further improvement of metal phosphides catalysts in DMR is therefore highly desirable to develop a new catalyst which is resistant to coking for the atmospheric DMR reaction.

Considering that the combination of the strong ability for dissociating methane of metallic Ni and strong ability for cracking carbon dioxide of Mo_2_C has exhibited excellent catalytic performance and anti-coking property, as a consequence, the system of metallic nickel and transition metal phosphide is worth taking into account for the DMR process. Herein, we developed a new metallic nickel–nickel phosphides catalyst system for DMR to cope with the inherent coking problem of Ni-based catalysts. Through incipient wetness impregnation method, we prepared a series of catalysts with different Ni/P molar ratio supported on the alumina. The catalysts were tested in DMR process at 700 °C and atmospheric pressure. At the same time, we attempted to clarify the effect of the addition of P on the catalyst properties and coke resistance in the DMR reaction.

## 2. Experimental

### 2.1. Catalyst Preparation

The Ni-based catalysts were prepared by an incipient wetness impregnation method with an aqueous solution of nickel nitrate hexahydrate (Ni(NO_3_)_2_·6H_2_O, AR, Sinopharm Chemical Reagent Co., Ltd., Shanghai, China). The impregnated samples were dried at 110 °C overnight, and then calcined at 800 °C in air for 4 h. P was introduced in the catalysts by co-impregnation method with diammonium hydrogen phosphate ((NH_4_)_2_HPO_4_, AR, Shanghai Aladdin Biochemical Technology Co., Ltd., Shanghai, China) as a precursor. Citric acid (AR, Sinopharm Chemical Reagent Co., Ltd., Shanghai, China) was employed as complexing agent with the same mole of nickel to avoid the generation of Ni-PO_4_ precipitation. Before dry methane reforming measurement, the catalysts were reduced at 900 °C for 2 h with pure H_2_ flow. The loading content of Ni was fixed at 10 wt.% (m_Ni_/m_catalyst_), and P-free catalyst was denoted as Ni/Al_2_O_3_ while P-containing catalysts were denoted as NiP-x/Al_2_O_3_, where the x is the nominal atomic ratio of Ni:P.

### 2.2. Catalyst Characterization

Transmission electron microscopy (TEM) and high-resolution transition electron microscopy (HRTEM) images were collected on a JEM 2100 transmission electron microscope (JEOL Ltd., Tokyo, Japan) and Tecnai F20 transmission electron microscope (FEI Company, Hillsboro, OR, USA) operating at 200 kV, respectively. High-angle annular dark field (HAADF) images in the STEM mode of the reduced catalysts were taken on a Talos F200× transmission electron microscope (Thermo Fisher Scientific, Waltham, MA, USA). The powder samples were dispersed in ethanol by ultrasonic and deposited on a copper grid to dry before measurement. At least ten representative images were taken for each sample. In order to obtain statistically reliable information, the size of at least 400 particles was measured.

X-ray diffraction (XRD) analyses of the bulk and crystalline structures of the catalysts were performed using a D8 ADVANCE X-ray diffractometer (Bruker Corporation, Karlsruhe, Germany) with Cu Kα radiation. The scanned diffraction peaks were in the 2θ range between 10 and 90° with a scan speed of 5.7°·min^−1^. The assignment of the various crystalline phases was based on the ICSD diffraction file cards.

X-ray photoelectron spectroscopy (XPS) analyses of the reduced catalysts were performed on an X-ray photoelectron spectrometer (AXIS SUPRA, Shimadzu Corporation, Kyoto, Japan). The binding energies (BE) of Ni 2p and P 2p core-levels were recorded. The binding energy of the C 1s peak at 284.8 eV was taken as charge correction standard.

Elemental analyses of the reduced catalysts were performed by inductively coupled plasma optical emission spectrometry (ICP-OES) on a SPECTRO ARCOS II instrument (SPECTRO company, Kleve, Germany) to determine the elemental content and actual Ni/P atomic ratio.

Nitrogen adsorption–desorption measurements of the reduced catalysts were carried out on an ASAP 2460 apparatus (Micromeritics company, USA). Before the measurements, ∼0.2 g catalysts were degassed under a flow of N_2_ at 200 °C for 12 h. Specific surface areas (SSA) were calculated based on the Brunauer–Emmett–Teller (BET) equation at P/Po values in the range of 0.05–0.25. Pore sizes were determined by the Barrett–Joyner–Halenda (BJH) method using desorption isotherms.

Thermogravimetric analyses (TG) were performed on Diamond TG/DTA equipment from 25 to 800 °C at a heating rate of 10 °C·min^−1^ under air flow in order to quantify the carbon deposition formed on surface of the spent catalysts.

H_2_ temperature programmed reduction (H_2_-TPR) measurements of calcined catalysts were carried out with a VDSorb-91i apparatus (Quzhou Vodo Instrument Co., Ltd., Quzhou, China). The 0.05 g powder samples were pretreated at 200 °C for 30 min under N_2_ flow at a rate of 30 mL·min^−1^ and then cooled down to 50 °C. The samples were subsequently reduced by increasing the temperature to 1000 °C at a ramp rate of 10 °C·min^−1^ in a stream of 8% H_2_/Ar flow with a rate of 30 mL·min^−1^. The H_2_ consumption was determined with a thermal conductivity detector (TCD). A cooling trap placed between the sample and the detector retained the water formed during the reduction process.

Temperature programmed desorption of NH_3_ (NH_3_-TPD) analyses of the reduced catalysts were performed on an AutoChem II 2920 chemisorption apparatus (Micromeritics company, USA) in order to investigate the acidity of the catalysts. Prior to the analyses, ~0.05 g samples were pretreated at 300 °C for 60 min under He flow at rate of 30 mL·min^−1^ and cooled down to 50 °C. Next, the samples were ammonia-saturated in a stream of 10% NH_3_/He flow at rate of 30 mL·min^−1^ for 60 min and then, the He was introduced for 30 min to remove the physical adsorbed NH_3_. Finally, the samples were heated to 900 °C at a ramp rate of 10 °C·min^−1^ under He flow.

In-situ CO_2_ adsorption diffuse reflectance infrared Fourier transform spectroscopy (DRIFTS) analysis were recorded by a Thermo Nicolet 6700 infrared spectrometer (Thermo Fisher Scientific, Waltham, MA, USA) to obtain information about the plausible DMR reaction mechanism. The catalysts were initially reduced by H_2_ and then purged with nitrogen at 300 °C for 30 min. After cooling down to 50 °C, 10 vol.% CO_2_/N_2_ mixture gas was introduced to the system. The spectral drift spectrum was recorded every 25 °C from 50–440 °C. Note that all spectra were recorded after 20 min exposure time when steady state conditions were reached.

### 2.3. Catalyst Performance Tests

The dry reforming of methane was carried out in a fixed-bed vertical quartz reactor (inner diameter = 8 mm) at atmospheric pressure. The reactor operated in a down flow mode and heated by a tubular resistance furnace. The bed of powder catalyst (100 mg, 20–40 mesh) was held in position by quartz wool and the thermocouple was placed near the catalyst bed tightly to measure the reaction temperature.

Before the DMR reaction tests, the catalysts were in-situ reduced at corresponding temperature at heating rate of 5 °C·min^−1^ for 2 h under H_2_ flow at a rate of 100 mL·min^−1^. Later, the reaction temperature was ramped to 700 °C and maintained for different hours. The feed gas consisted of CH_4_ (33.3 vol.%) and CO_2_ (33.3 vol.%) diluted with N_2_. Nitrogen was employed as the internal standard gas. The purity of the gases used in this work was 99.999 vol%. The outlet gas was analyzed using an online gas chromatograph (GC-7960 plus, Tengzhou Allen Analytical Instruments Co., Ltd., Tengzhou, China) equipped with a thermal conductivity detector and TDX-01 column.

## 3. Results and Discussion

### 3.1. Characterization of Fresh Catalysts

Element analysis of reduced catalysts, performed by ICP-OES, are displayed in Table 1. The measured contents of nickel and phosphorus roughly match with their nominal values, and the actual Ni/P molar ratios are basically in line with their expected values. This implies that there was no visible mass loss for phosphorus during reduction process at high temperature through PH_3_ gas.

The textural properties of the reduced catalysts, characterized by nitrogen adsorption–desorption measurements are summarized in Table 1 and the N_2_ adsorption–desorption isotherms of Ni/Al_2_O_3_ and NiP-x/Al_2_O_3_ samples are shown in Figure 1b. Based on the IUPAC classification, the nitrogen isotherms can be classified as type IV, which is typical for mesoporous material [38]. With the change of P content, no significant change in the pore volume and average pore diameter of the catalysts was observed.

X-ray diffraction technology was used to study the crystal structure of reduced catalysts. The XRD patterns of reduced NiP-x/Al_2_O_3_ catalysts are presented in Figure 1a. Characteristic peaks at 2θ = 31.94°, 37.60°, 39.49°, 45.86°, 67.03° can be assigned to the crystalline phase of Al_2_O_3_ (JCPDS 10-0425) support. Characteristic peaks at 2θ of 44.49°, 51.85°, 76.38° can be assigned to the crystallographic planes of metallic nickel phase (JCPDS 65-2865). Major peaks broadening indicates high dispersion of nickel phase on the alumina support. After the introduction of phosphorus, the further broadening of characteristic peaks of metallic nickel implies the better dispersion of nickel species. Besides, evident peak at 2θ of 48.96° along with broaden peaks at 2θ of 41.78°, 47.65°, can be assigned to the crystallographic planes of Ni_12_P_5_ phase (JCPDS 22-1190) and Ni_5_P_2_ phase (JCPDS 17-0225), respectively, are observed over the NiP-2/Al_2_O_3_ catalyst, which implies the co-existence of metallic nickel and nickel phosphides. Due to the relative lower phosphorus content and partial peaks overlapping between Ni_12_P_5_, Ni_5_P_2_, and alumina, the characteristic peaks of nickel phosphides are hard to distinguish over NiP-4/Al_2_O_3_ and NiP-3/Al_2_O_3_ catalysts.

The morphologies of the samples are shown in Figure 2. For the P-containing catalysts, nanoparticles are evenly dispersed on the surface of Al_2_O_3_ (Figure 2b–d). The nickel particle size distributions of γ-Al_2_O_3_ supported catalysts are summarized in Table 1. Compared to Ni/Al_2_O_3_ (Appendix A), the NiP-x/Al_2_O_3_ show the smaller nickel particle size, which coincides with the XRD results. Furthermore, the EDS elemental mapping analyses indicate that the element distribution of P is always consistent with nickel species, which indicates that nickel species are prone to enrich at the phosphorus existence place (Figure 2j). Based on HRTEM images (Figure 2e–g), the lattice fringes spacing of individual particle is estimated for its identification. For all the reduced catalysts, the lattice fringes spacing of 0.201 nm, which corresponds well to the characteristic (1 0 1) plane of Ni species, is observed obviously on the nickel species nanoparticles. What is more, the lattice fringes spacing of 0.186 nm and 0.198 nm, ascribed to the (3 1 2) crystal plane of Ni_12_P_5_ and the (2 2 10) crystal plane of Ni_5_P_2_, respectively, are observed at the reduced NiP-2/Al_2_O_3_ catalyst, (inset in Figure 2e–g). These results further confirm the co-existence of metallic nickel and nickel phosphides, which is in agreement with the results from the XRD of the reduced catalysts.

XPS analyses for the reduced catalysts were performed to investigate the surface valence states and electronic effects. Before the measurements, the catalysts have been passivated in order to be transferred from the quart reactor to the XPS cell. The Ni 2p XPS spectrum of reduced catalysts is illustrated in Figure 3a. The broad enveloping peak of the Ni 2p signal can be deconvoluted in several separate overlapping doublets indicating that Ni exists in different chemical states:-The peak at binding energy (BE) of 852.1–852.3 eV can be attributed to Ni^0^ species which may be involved in metallic nickel [6,39].-The peak at binding energy of 853.1–853.2 eV can be attributed to Ni^δ+^ (0 < δ < 2) which may be involved in nickel phosphides species [40,41].-The peak at binding energy of 855.3–856.5 eV can be attributed to Ni^2+^ which may be involved in Ni^2+^ in nickel phosphates or nickel aluminates species [39,42].

The presence of Ni^2+^ species can be explained by the passivation step which oxidized the surface Ni. However, it cannot be totally excluded that the presence of oxides can be due to incomplete reduction [43].

As shown in Figure 3a, Ni^0^ and Ni^2+^ species along with their corresponding satellites are identified in the Ni 2p spectra of reduced Ni/Al_2_O_3_ samples. For the reduced P presence catalysts, the peaks at 853.1–853.2 eV, attributed to Ni^δ+^ species, are confirmed, which indicates the existence of nickel phosphides phase on the reduced NiP-_X_/Al_2_O_3_ catalysts. Table 2 displays the nickel species distribution obtained by XPS. The result indicates a considerable percentage of Ni^2+^ species (70.9%) and a low percentage of Ni^0^ species (29.1%) on the Ni/Al_2_O_3_ catalyst. The percentage of Ni^δ+^ species increases with the increasing P content, which are obviously observed on the P presence catalysts. Moreover, the presence of P increases the percentage of Ni^0^ and Ni^δ+^ while decreases Ni^2+^ species, which are observed on the reduced NiP-3/Al_2_O_3_ and NiP-2/Al_2_O_3_ catalysts. This may imply that the presence of P improves the reduction capacity of Ni/Al_2_O_3_ catalyst.

In P 2p spectra, as shown in Figure 3b, the broad enveloping peak of the P 2p signal can be deconvoluted in two separate overlapping doublets, indicating that P exists in different chemical states. The peak at binding energy of 129.8–129.9 eV can be assigned to P^δ−^ which may be involved in nickel phosphides species [40,41]; the peaks at binding energy of 133.5–133.7eV can be attributed to P^5+^ and may be involved in phosphate species [35,41]. The presence of phosphates species may be due to the oxidation during passivation process and/or incomplete phase transformation during reduction process. The obvious peaks at 129.8–129.9 eV, attributed to Ni^δ+^ species, are confirmed on all the reduced NiP-_X_/Al_2_O_3_ catalysts, which strongly suggests the existence of nickel phosphides phase.

H_2_-TPR measurements were employed to investigate the interaction between Ni, P species, and alumina support of the calcined catalysts. Unsupported bulk NiO is known to be reduced to metallic Ni in the 200–450 °C range [33,44]. However, as is shown in Figure 4, the main reduction peak of nickel shifted to 700 to 900 °C temperature range, which may be due to the strong interaction between NiO and alumina support. These strong NiO-Al_2_O_3_ interactions are caused by the dissolution and incorporation of Al^3+^ ions in NiO crystallites [44,45]. A new shoulder peak at temperature range of 900–1000 °C, corresponding to the formation of AlPO_4_ [38,46], along with an obvious shift to higher temperature of the main nickel species peaks are observed for the P-containing catalysts, which implies that the presence of P results in stronger metal–support interaction between nickel and alumina support.

NH_3_-TPD was performed to investigate the acidic property of the samples. Figure 5 shows the NH_3_-TPD profiles of Al_2_O_3_ supported catalysts reduced at 900 °C. γ-Al_2_O_3_ support has been also tested in order to better define the effect of phosphorous on surface acidity. For the reference alumina support, three main NH_3_ desorption peaks centered at ~123 °C, ~259 °C, and ~587 °C, which are associated with weak, medium, and strong acid sites, respectively, are identified. It is noted that evident decrease of peak area of Ni/Al_2_O_3_ NH_3_-TPD profiles is observed in comparison with the reference Al_2_O_3_, which indicates that nickel could weaken the surface acidity. What is more, obvious peak area decreases for the phosphorus presence catalysts, which suggests that phosphorus could decrease the surface acidity on Ni/Al_2_O_3_ catalyst.

### 3.2. Catalytic Performance

Dry methane reforming results with time on stream for 50 h, which were performed over reduced catalysts at 700 °C, CH_4_:CO_2_:N_2_ = 1:1:1, the WHSV = 18,000 mL gcat^−1^ h^−1^, ambient atmosphere, are shown in Figure 6. As is indicated, the Ni/Al_2_O_3_ catalyst exhibits excellent catalytic performance. The methane and carbon dioxide conversion are as high as 80% and 87%, which is close to the equilibrium conversion. The higher CO_2_ conversion than methane conversion may be due to the simultaneous occurrence of the reverse water gas shift reaction (CO_2_ + H_2_ = H_2_O + CO) [47], which is also the reason for lower H_2_/CO ratio than stoichiometric amount. A slight decrease of activity is observed for the P presence catalysts in dry methane reforming. This decrease of activity is probably partially due to both the partial coverage of metallic nickel by phosphorus species and less metallic nickel as the existence of nickel phosphides species. Note that no further activity decrease is observed with the increasing phosphorus content at our study range.

For further study, the stability of Ni/Al_2_O_3_ and NiP-x/Al_2_O_3_ catalysts was investigated for at least 50 h and the results are shown in Figure 7. It is indicated that an activity degradation is observed within the 50 h of time on stream over Ni/Al_2_O_3_ catalyst. The CH_4_ and CO_2_ conversion dropped from 80% and 87% to 76% and 84%, respectively. The activity degradation may be due to the surface carbon deposition and/or particle sintering, which are common reasons for activity degradation over nickel-based catalysts during dry methane reforming process. It is worth pointing out that all the P presence catalysts exhibited better stability than Ni/Al_2_O_3_ catalyst during dry methane reforming. No visible activity loss was observed within 50 h of time on stream.

### 3.3. Characterization of Spent Catalysts

TG analyses were performed on the spent catalysts after 50 h DMR reaction to determine the amount of carbon deposition. As is shown in Figure 8, all the spent catalysts exhibit several weight changes in temperature range of 30–800 °C. The first weight loss below 200 °C can be associated with the removal of physically adsorbed H_2_O, while the tiny weight gain between 240–360 °C can be attributed to oxidation of the metallic Ni to its corresponding oxides [48]. The second weight loss at 450–700 °C can be ascribed to the oxidation of graphitic carbon deposited on the catalyst during the DMR reaction [49,50]. No evident weight loss is observed in the temperature range of 240–360 °C, attributed to the oxidation of surface amorphous carbon [49]. This implies that the generation graphitic carbon is the main reason for carbon deposition, which coincides with the literature. Note that obvious less weight loss at 450–700 °C is observed for the phosphorus presence catalysts and the content of carbon deposition decreases in the following order: Ni/Al_2_O_3_ (34.8%) > NiP-4/Al_2_O_3_ (21.8%) > NiP-3/Al_2_O_3_ (17.5%) > NiP-2/Al_2_O_3_ (2.7%), which is in line with the intensity trend of the graphitic carbon diffraction peak in the XRD patterns (Figure 9). The result strongly implies that the existence of P species results in better resistance of carbon deposition over the catalysts during dry methane reforming process. The enhancement of carbon deposition resistance of P presence catalysts might be one of the reasons for better stability in dry methane reforming. The tendency indicates that when the amount of P increases, the degree of carbon deposited on the surface of the catalyst decreases.

The XRD patterns of spent Al_2_O_3_ supported catalysts are shown in Figure 9. Compared with the fresh catalysts, the relative sharp peaks shape of metallic nickel and nickel phosphides species indicates the growth of crystalline size during dry methane reforming at high temperature. Phosphorus-contained catalysts indicate relative wide peaks of nickel species, implying better anti-sintering properties compared to Ni/Al_2_O_3_ catalyst. Characteristic peak at 2θ of 26.2°, attributed to graphitic carbon due to the carbon deposition during dry methane reforming process, is detected over all the catalysts. Note that relative peak intensity of NiP-x/Al_2_O_3_, especially for the NiP-3/Al_2_O_3_ and NiP-2/Al_2_O_3_, decreases obviously, indicating less carbon deposition, which coincides with the TG result.

TEM images and particle size distributions of the spent catalysts are shown in Appendix A. The particle size distributions of catalysts before and after the DMR reaction are summarized in Table 3. As is shown in the TEM images, obvious filamentous carbon species are observed encapsulating on Ni/Al_2_O_3_ catalyst, indicating the severe carbon deposition, which is also confirmed by TG profiles and XRD patterns. As is depicted in Table 3, a small enhancement of particle sizes of nickel species is observed for all the spent catalysts, which coincides with the XRD result. Note that the enhancing range of nickel species particle size is obviously restrained for the phosphorus presence catalysts. This result further implies that the presence of phosphorus increases the anti-sintering properties of nickel catalyst.

### 3.4. DRIFTS Analysis

In-situ CO_2_ adsorption diffuse reflectance infrared Fourier transform spectroscopy (DRIFTS) analyses were carried out to obtain information about the plausible DMR reaction mechanism and Appendix A, Figure 10 recorded spectra. The bands between 2250–2450 cm^−1^ can be assigned to CO_2_ species, whose relative concentration changes with temperature. At 50 °C, the band at 2344 cm^−1^, together with two bands localized at 3700 and 3600 cm^−1^, are observed on the spectra for both catalysts, and can be attributed to the gas phase CO_2_ [11,51,52,53,54]. At higher temperature, new bands at 2386, 2377, 2332, 2319, 2309, 2294, and 2284 cm^−1^ are observed, and can be attributed to the chemisorbed species of CO_2_ on the Lewis acid sites of alumina support [55,56]. Note that obvious depletion of CO_2_ overtones at bands of 2344, 3600, and 3700 cm^−1^ [51,53], along with the enhanced intensity of the bands, belong to adsorbed CO_2_ species and are observed with the increasing temperature. This result implies that higher temperature is favorable for the chemisorption of CO_2_ over the catalysts.

The bands between 1800–1200 cm^−1^ indicate the formation of adsorbed C-containing species on the catalyst surface during reaction, whose relative concentration changes with temperature. At 50 °C, both spectra show bands at 1652 and 1430 cm^−1^ that can be assigned to O–C–O stretching (ν_OCO_, both symmetric and asymmetric) of bicarbonate species [51,53]. These species are formed by protonation of carbonate species with alkaline hydroxyl OH* active species. The O-H stretching bands of these bicarbonates is also observed at 3725 and 3625 cm^−1^. Moreover, the band at 1562 cm^−1^ can be attributed to the asymmetric O–C–O stretching which is the characteristic band of formate species adsorbed on Al_2_O_3_. The formate species, formed by the further evolution of bicarbonate species, are one of the possible reaction intermediates, and could further yield CO. Besides, the tiny bands at 2931 and 2862 cm^−1^, related to the CH_x_ stretching mode of formats, confirms the presence of these species. The formate species, created by the further evolution of bicarbonate species, are one of the possible reaction intermediates, and could further yield CO. The existence of gas phase CO is proved by the tiny band at around 2230 cm^−1^ at higher temperature. With the increase of temperatures, evidence the depletion of bicarbonates and the formation of formates are observed for both catalysts. The evolution of bands with temperature suggests that CO_2_ proceeds via formation of carbonyl species, mainly arising from intermediate formates decomposition to CO during dry methane reforming.

Note that two new bands at 3780 and 3674 cm^−1^ are observed over NiP-2/Al_2_O_3_ catalyst, which can be attribute to the hydroxyl groups of AlO-H and PO-H species, respectively, belonging to the AlPO_4_ phase [57,58,59]. The existence of AlPO_4_ phase is also confirmed by H_2_-TPR and XPS spectra. These hydroxyl groups might benefit for the adsorption of CO_2_ on catalyst during reaction.

Compared to Ni/Al_2_O_3_, the spectrum of NiP-2/Al_2_O_3_ presents evident higher intensity of the bands of adsorbed CO_2_ species, which implies the higher CO_2_ adsorption ability of NiP-2/Al_2_O_3_ catalyst. Additionally, lower bicarbonates bands at 1652 and 1430 cm^−1^ and higher formates species bands at 1562, 2931, and 2862 cm^−1^ over NiP-2/Al_2_O_3_ catalyst are also observed. This result indicates that phosphorus presence catalysts own better activation ability of CO_2_. The produced bicarbonates species are more easily further converted to formate intermediates and active O* species, which favors the elimination of the carbon deposition.

## 4. Discussion

It is known that nickel catalyst shows excellent catalytic activity in dry methane reforming. However, due to the relative higher CH_4_ dissociation ability of nickel catalysts, serve carbon deposition restricts its application. Previous reports indicated that the transition metal phosphides expressed excellent catalytic performance and carbon deposition resistance for dry methane reforming. Nevertheless, due to the relative higher CO_2_ dissociation ability of transition metal phosphides, oxidation of phosphides to phosphate phase was also observed at high temperature and GHSV condition during dry methane reforming. In this work, the strong CH_4_ dissociation ability of metallic nickel and excellent CO_2_ dissociation ability of nickel phosphides were combined to obtain better carbon deposition resistance.

A series of NiP-x/Al_2_O_3_ catalysts containing different ratio of metallic nickel to nickel phosphides were prepared by varying Ni/P molar ratio of 4, 3, 2 through a co-impregnation method. The obtained results reveal an evident influence of phosphorus content on the catalyst structure and the catalytic performances in dry methane reforming. ICP results indicate that the measured contents of nickel and phosphorus roughly match with their nominal values, which implies no visible mass loss of phosphorus during reduction at high temperatures. XRD and TEM suggest that nickel is highly dispersed on the alumina surface support and the P presence catalysts exhibit smaller nickel particle size. The co-existence of metallic nickel and nickel phosphides phase is confirmed by XRD, XPS, and HRTEM images. Moreover, nickel species are prone to enrich the phosphorus existence place, which is confirmed by element mapping analysis. H_2_-TPR highlights strong interaction between nickel and alumina support for non-promoted samples after calcination. The addition of phosphorus is supposed to reinforce this interaction between nickel and alumina support.

As expected, Ni/Al_2_O_3_ catalyst shows good activity while having a slight deactivation after 50 h of time on stream in dry methane reforming. XRD data of the spent Ni/Al_2_O_3_ sample reveals the enhancement of nickel crystallite size. Moreover, severe carbon deposition and nickel particle size enhancement are detected by TG, XRD, and TEM analyses. The increase of nickel particle size and carbon deposition should be the reasons for the slight deactivation. The phosphorus containing catalysts exhibit slight lower activity while showing better stability in dry methane reforming. Obvious lower carbon deposition and particle size increase are detected by TG and TEM analyses over the spent P-containing catalysts. In addition, in-situ CO_2_ adsorption DRIFTS analysis gives information that the phosphorus containing catalysts own evident higher CO_2_ adsorption and dissociation ability, which coincides with the publications [36,41]. The better CO_2_ dissociation ability and anti-sintering property should be responsible for the better anti-coking and stability of the phosphorus containing catalysts. Note that the carbon deposition is strongly influenced by the phosphorus content. The NiP-2/Al_2_O_3_ catalyst exhibits best resistance of carbon deposition, which may be due to the better balance between CH_4_ dissociation and CO_2_ dissociation over the catalyst during dry methane reforming.

## 5. Conclusions

A series of NiP-x/Al_2_O_3_ catalysts containing different ratios of metallic nickel to nickel phosphides, prepared by varying Ni/P molar ratio of 4, 3, 2 through a co-impregnation method, were employed to investigate the synergistic effect of metallic nickel–nickel phosphides in dry methane reforming reaction. Good dispersion of nickel particles and strong nickel-alumina interaction is confirmed by a series of characterizations over the P-free catalyst. In addition, the Ni/Al_2_O_3_ catalyst indicates good activity but severe carbon deposition in dry methane reforming reaction.

The presence of phosphorus increases nickel dispersion as well as the interaction between nickel and alumina support, which results in smaller nickel particles. The co-existence of metallic nickel and nickel phosphides species is confirmed. Moreover, compared to the Ni/Al_2_O_3_ catalyst, phosphorus contained catalysts exhibit higher CO_2_ dissociation ability. Due to the relative stronger CO_2_ dissociation ability, NiP-x/Al_2_O_3_ catalysts indicate obvious higher resistance of carbon deposition. Furthermore, because of good balance between CH_4_ dissociation and CO_2_ dissociation, NiP-2/Al_2_O_3_ catalyst exhibits best resistance of carbon deposition, and few carbon depositions were formed after 50 h of dry methane reforming.

## Figures and Tables

**Figure 1 materials-15-03044-f001:**
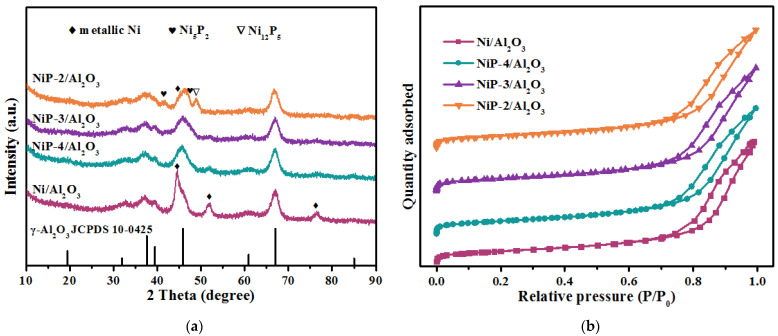
(**a**) XRD patterns and (**b**) N_2_ adsorption–desorption isotherms of reduced Ni/Al_2_O_3_ and NiP-x/Al_2_O_3_ catalysts.

**Figure 2 materials-15-03044-f002:**
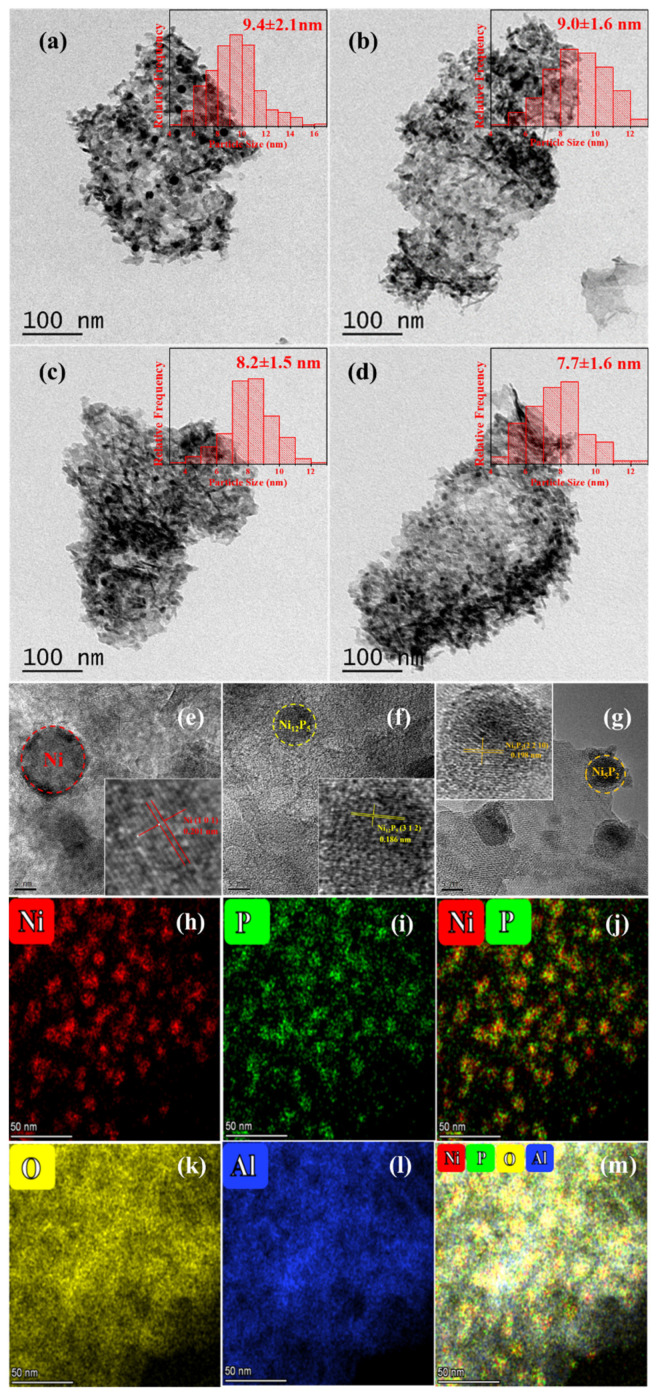
TEM images and corresponding size distributions of reduced (**a**) Ni/Al_2_O_3_; (**b**) NiP-4/Al_2_O_3_; (**c**) NiP-3/Al_2_O_3_; (**d**) NiP-2/Al_2_O_3_; (**e**–**g**) HRTEM image (the inset is the magnified image of Ni, Ni_12_P_5_ and Ni_5_P_2_) for reduced NiP-2/Al_2_O_3_ catalyst; and (**h**–**m**) elemental mapping results for reduced NiP-2/Al_2_O_3_ catalyst.

**Figure 3 materials-15-03044-f003:**
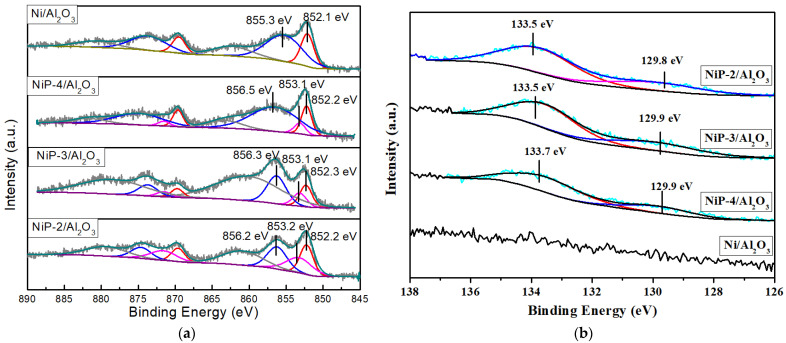
(**a**) Ni 2p and (**b**) P 2p XPS spectra of reduced Ni/Al_2_O_3_ and NiP-x/Al_2_O_3_ catalysts.

**Figure 4 materials-15-03044-f004:**
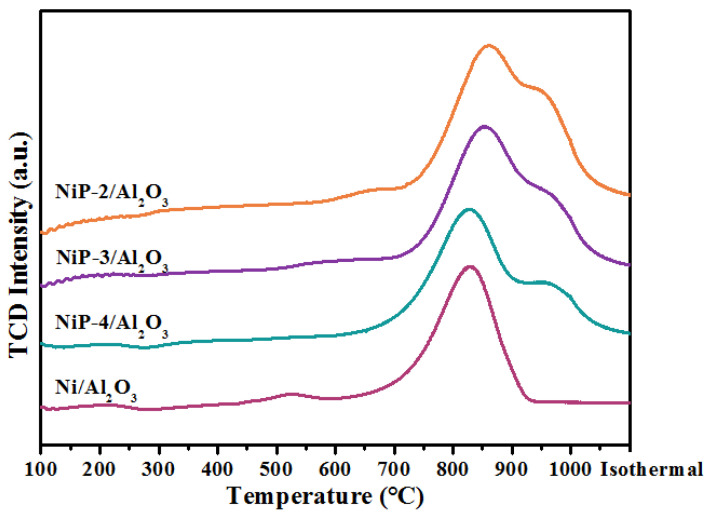
H_2_-TPR profiles of reduced Ni/Al_2_O_3_ and NiP-x/Al_2_O_3_ catalysts.

**Figure 5 materials-15-03044-f005:**
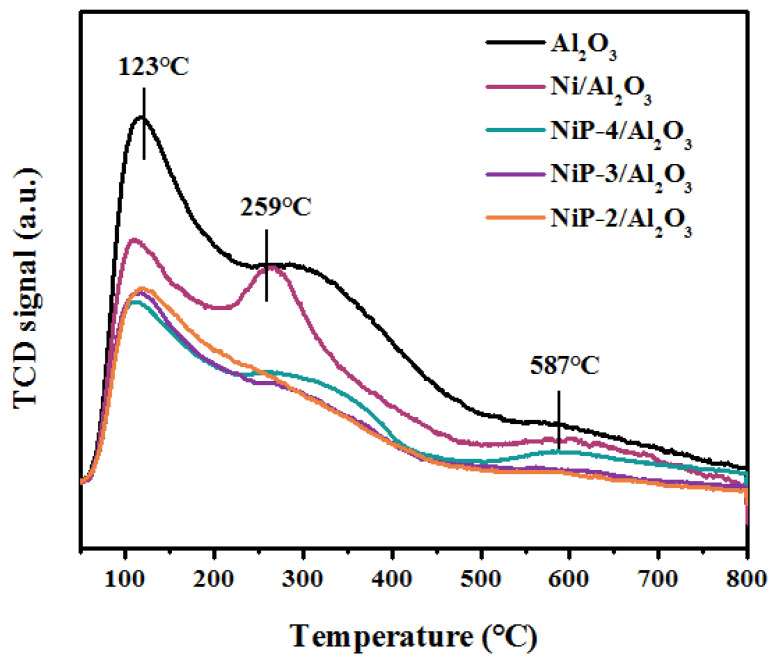
NH_3_-TPD profiles of reduced Ni/Al_2_O_3_ and NiP-x/Al_2_O_3_ catalysts.

**Figure 6 materials-15-03044-f006:**
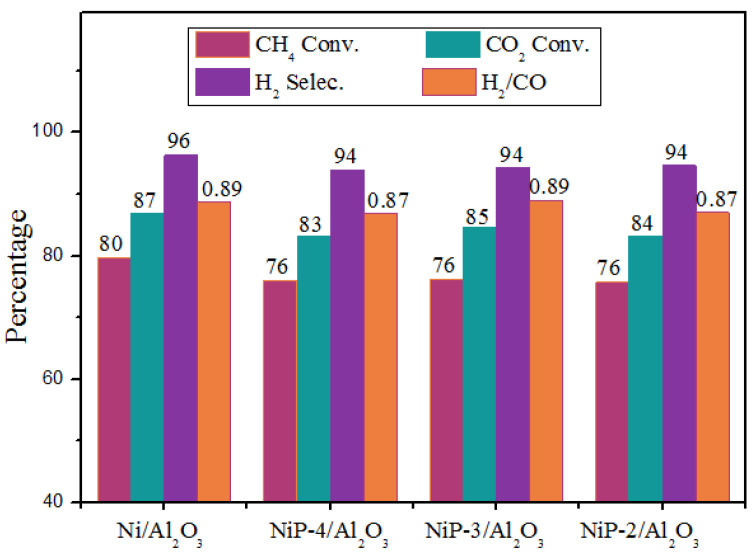
Catalytic performance of Ni/Al_2_O_3_ and NiP-x/Al_2_O_3_ catalysts in dry methane reforming.

**Figure 7 materials-15-03044-f007:**
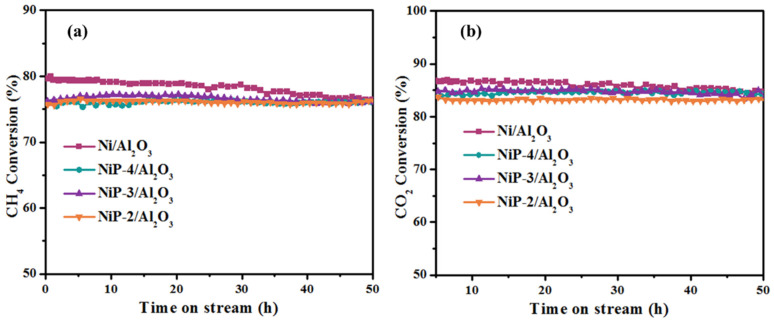
(**a**) CH_4_ and (**b**) CO_2_ conversion as a function of time on stream over Ni/Al_2_O_3_ and NiP-x/Al_2_O_3_ catalysts.

**Figure 8 materials-15-03044-f008:**
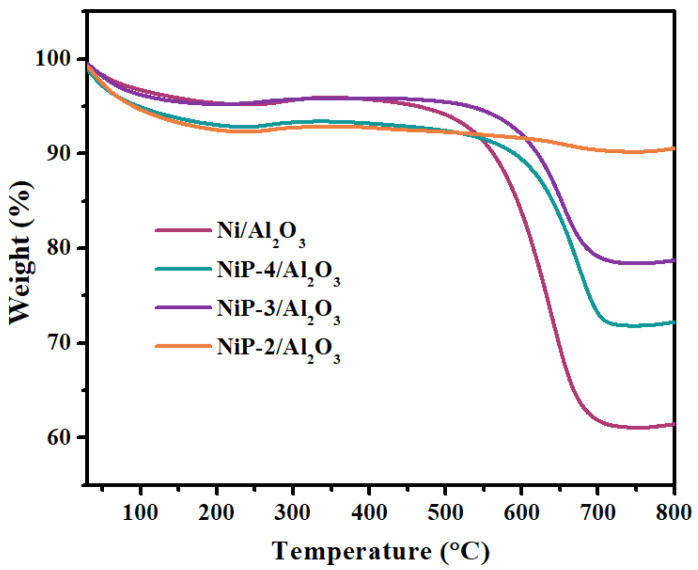
TG profiles of spent Ni/Al_2_O_3_ and NiP-x/Al_2_O_3_ catalysts.

**Figure 9 materials-15-03044-f009:**
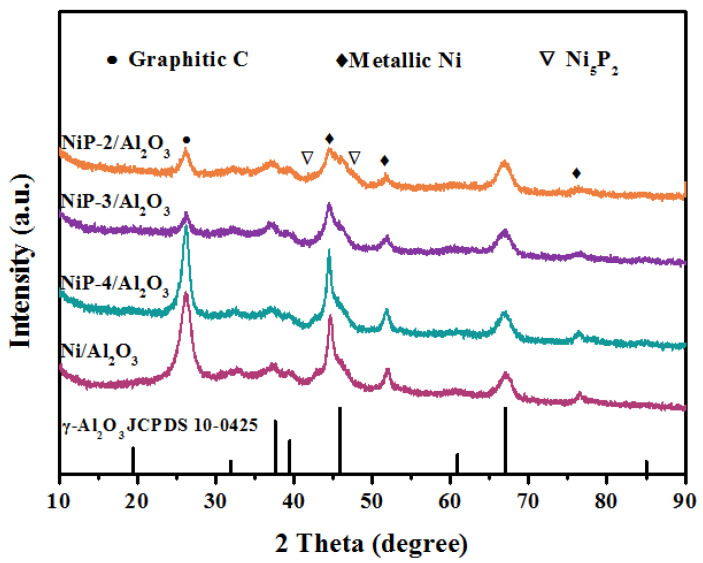
XRD patterns of spent Ni/Al_2_O_3_ and NiP-x/Al_2_O_3_ catalysts.

**Figure 10 materials-15-03044-f010:**
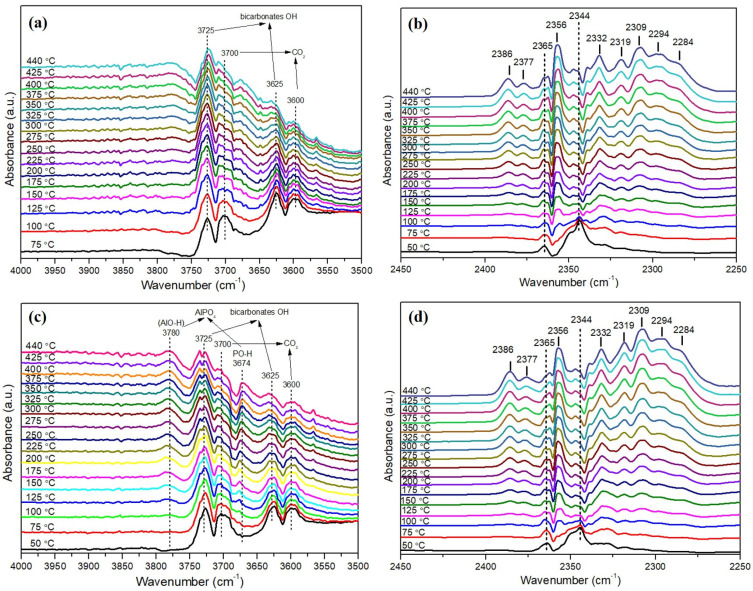
Partial DRIFT spectra of CO_2_ adsorbed at different temperatures over Ni/Al_2_O_3_ with the wavelength 3500–4000 cm^−1^ (**a**), 2250–2450 cm^−1^ (**b**), and over NiP-2/Al_2_O_3_ catalyst with the wavelength 3500–4000 cm^−1^ (**c**), 2250–2450 cm^−1^ (**d**).

**Table 1 materials-15-03044-t001:** Chemical composition, textural properties, and particle size distribution of reduced Al_2_O_3_ supported catalysts.

Catalyst	Chemical Composition			Textural Properties			Size ^b^ (nm)
	Ni ^a^ (wt.%)	P ^a^ (wt.%)	Ni/P Molar Ratio	S_BET_ (m^2^/g)	V_pore_ (cm^3^/g)	D_pore_ (nm)	
**Ni/Al_2_O_3_**	9.56	-	-	156	0.66	13.33	9.4 ± 2.1
**NiP-4/Al_2_O_3_**	9.81	1.30	3.97	171	0.68	13.50	9.0 ± 1.6
**NiP-3/Al_2_O_3_**	8.56	1.43	3.17	167	0.66	12.61	8.2 ± 1.5
**NiP-2/Al_2_O_3_**	10.0	2.54	2.04	162	0.64	13.19	7.7 ± 1.6

^a^ Determined from ICP-OES. ^b^ Estimated by TEM.

**Table 2 materials-15-03044-t002:** Ni species distribution (Ni^X+^/Ni ratio) of reduced Ni/Al_2_O_3_ and NiP-x/Al_2_O_3_ catalysts.

Catalyst		Nickel Species Distribution		
		Ni^0^	Ni^δ^^+^	Ni^2+^
Ni/Al_2_O_3_	Position (eV)	852.1	-	855.3
	Species distribution	29.1	0	70.9
NiP-4/Al_2_O_3_	Position (eV)	852.2	853.1	856.5
	Species distribution	18.8	7.7	73.5
NiP-3/Al_2_O_3_	Position (eV)	852.3	853.2	856.3
	Species distribution	26.8	18.5	54.7
NiP-2/Al_2_O_3_	Position (eV)	852.2	853.2	856.2
	Species distribution	32.0	32.4	35.6

**Table 3 materials-15-03044-t003:** Particle size distributions of fresh and spent Ni/Al_2_O_3_ and NiP-x/Al_2_O_3_ catalysts.

Catalyst	Particle Size Distribution (nm)		Particle Size Enhancing Range (%)
	Fresh Catalysts	Spent Catalysts	
Ni/Al_2_O_3_	9.4 ± 2.1	10.3 ± 2.5	9.6
NiP-4/Al_2_O_3_	9.0 ± 1.6	9.5 ± 1.5	5.6
NiP-3/Al_2_O_3_	8.2 ± 1.5	8.5 ± 1.4	3.7
NiP-2/Al_2_O_3_	7.7 ± 1.6	8.4 ± 1.8	9.1

## Data Availability

Not applicable.

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
