# Peer review of "Improving Anti-Coking Properties of Ni/Al2O3 Catalysts via Synergistic Effect of Metallic Nickel and Nickel Phosphides in Dry Methane Reforming"

_materials, 2022, doi:10.3390/ma15093044_

Round 1

Reviewer 1 Report

Dear authors, see the attached file.

Author Response

Response to Reviewer 1 Comments

Dear Reviewers,

Thank you for your time reviewing the manuscript and your very thorough comments on the article. We appreciate your clear and detailed feedbacks and hope that the following explanations could address all of your concerns. In the remainder of this letter, we discuss each of your comments along with our corresponding responses.

To facilitate this discussion, we first retype your comments in italic font and then present our responses to the comments in red.

Point 1: Table 1. The samples, containing phosphorus, have the same names (NiP-4/Al2O3).

Response 1:

Thanks for the kind comment of the reviewer, the editing error has been revised in Table 1 of the manuscript.

Point 2: Table 1. It should be round to tenths rather than hundredths in the Size column. It is very difficult to measure with such accuracy, which is indicated by you.

Response 2:

Thanks for the kind comment, we agree with the reviewer that the particle sizes measured from TEM pictures are not such precise. The particle size data should be round to tenths in Table 1, Figure 2 and Table 3.

Point 3: It is necessary to indicate the manufacturers and the purity of reagents in the section 2.1 “Catalyst preparation” and also of gases in the section 2.3 “Catalyst performance tests”

Response 3:

Thanks for your great suggestion on improving the accessibility of our manuscript. The manufacturers and the purity of the regents and gases have been added in section 2.1 “Catalyst preparation” and section 2.3 “Catalyst performance tests” of the manuscript.

Point 4: Lines 99-100.” The apparent crystallite size was determined using the Debye-Scherrer equation”. I could not find numerical data of this estimation.

Response 4:

Thanks for the kind comment of the reviewer, the crystallite size of nickel particles can be determined by the 2θ and FMHW from XRD patterns using the Debye-Scherrer equation. However, as the broaden diffraction peaks and the partial overlapping of peaks for metallic nickel and alumina support, the crystallite size data determined by the Debye-Scherrer equation may not be as reliable as the statistical results from TEM pictures. Thus we delete the crystallite size data in the submitted manuscript. The editing error has been revised in the revised manuscript.

Point 5: What do you think influences the type of the emerging phosphide phase (Ni2P, Ni12P5, Ni5P2)? Were all three mentioned phases found in each of the catalysts?

Response 5:

A very wide range of stoichiometries can be exhibited for nickel phosphides, there are at least eight nickel phosphide stoichiometries known—Ni3P, Ni5P2, Ni12P5, NiP2, Ni5P4, NiP, NiP2 andNiP3 (Chem. Soc. Rev., 2010, 39, 4388–4401). It’s easy to get the corresponding bulk nickel phosphide phase from the stoichiometric nickel/phosphrus precursor. However, for the supported catalyst, the nickel phosphide stoichiometries are influenced not only by the nickel/phosphrus precursor ratio, but also by the support as the possible interaction between nickel and/or phosphrus and support. For the NiP-x/Al2O3 catalysts in the manuscript, a considerable amount of AlPO4 phase formed, which evidently effect the stoichiometries of obtained nickel phosphides species. Because of the relative lower phosphorus content of NiP-4/Al2O3 and NiP-3/Al2O3 catalyst and the broaden shape of the XRD peaks, it is difficult to find the characteristic peaks of Ni2P, Ni12P5, Ni5P2 species.

Point 6: Lines 215-216 “The peak at binding energy of 855.3-856.2 eV can be attributed to Ni2+ which may be involved in Ni2+ in nickel phosphates or NiAl2O4 species”. You suggest the possible formation of NiAl2O4 phase. Can its formation affect the activity of catalysts? If yes, how?

Response 6:

Bulk NiO is known to be reduced to metallic Ni in the 200–450 °C ºC range (Catalysis Today 2014, 233, 46-52, Fuel Processing Technology 2010, 91, 185-193.), however, the H2-TPR curves indicate the main reduction peak of nickel is shift to 700 to 900 °C temperature range, which may be due to the strong interaction between NiO and alumina support. These strong NiO-Al2O3 interactions are caused by the dissolution and incorporation of Al3+ ions in NiO crystallites, which generates nickel aluminate species. The XRD indicated characteristic peaks at 2θ = 31.94°, 37.60°, 39.49°, 45.86°, 67.03°, assigned to the crystalline phase of Al2O3 (JCPDS 10-0425) support and characteristic peaks at 2θ of 44.49°, 51.85°, 76.38°, assigned to the crystallographic planes of metallic nickel phase (JCPDS 65-2865). As the broaden shape of the XRD peaks, it is hard to find the characteristic peaks belong to spinel structure NiAl2O4 species (JCPDS 78-1601). In addition, the characteristic peaks of some nickel aluminates such as Ni2Al18O29 (JCPDS 22-0451), NiAl32O49 (JCPDS 20-0777), NiAl26O40 (JCPDS 20-0776), are overlapped with Al2O3 (JCPDS 10-0425). So it’s difficult to distinguish the nickel aluminates species from alumina through the XRD patterns.

Thus, the characterization results show the generation of nickel aluminate species, but can not confirm the formation of NiAl2O4 species definitely. The description of the possible formation of NiAl2O4 phase in the manuscript is not accurate and revisions have been made in the manuscript (Line 252-253).

The formation of NiAl2O4 phase affects the activity of nickel catalysts in DMR. The NiAl2O4 spinel phase is generally considered as an inactive component for the CH4 dissociation. Whereas, the formation of NiAl2O4 phase implies the strong interaction between nickel and alumina, results in well dispersed of nickel, smaller and homogeneous nickel particles as well as the enhancement of resistance of sintering in dry methane reforming (Fuel 2014, 137, 155-163).  

Point 7: Lines 237-238. The peak at binding energy of 129.8-129.9 eV can be assigned to Pσ- may be involved in Ni phosphides species. I assume you mean Pδ-.

Response 7:

Thanks for the kind comment. The peak at binding energy of 129.8-129.9 eV can be assigned to Pδ-, not Pσ-. The editing error has been revised in the manuscript.

Point 8: Lines 238-239. “The peaks at binding energy of 133.5-133.7 eV can be attributed to P6+ may be involved in phosphate species”. P5+?

Response 8:

Thanks for the kind comment. The peak at binding energy of 133.5-133.7 can be assigned to P5+, not P6+. The editing error has been revised in the manuscript.

Point 9: Figure 3b. For NiP-4/Al2O3 the value of b.e. should be 129.9 eV, not 139.9 eV.

Response 9:

Thanks for the kind comment of the reviewer, the editing error has been revised in Figure3 b of the manuscript.

Point 10: Did inactive AlPO4 formed due to interaction between nickel phosphide and support (before and after 50 h of reaction)?

Response 10:

Based on the following characterization results, we can conclude that AlPO4 phase formed over the NiP-x/Al2O3 catalysts. For the H2-TPR curves, except the main reduction peaks at 700-900 °C temperature range, attributed to nickel species; a new shoulder peak at a temperature range of 900-1000 °C, corresponding to the formation of AlPO4 (Catalysis Today, 2018, 303, 100-105; Int. J. Hydrogen Energy, 2014, 39, 4909-4916), was observed over each P present catalysts. In addition, considerable peaks attributed to P5+ in phosphate were confirmed by P 2p XPS spectra, which indicated the existence of phosphate phase. Moreover, In-situ CO2 DRIFTS analysis showed two bands at 3780 cm-1 and 3674 cm-1 over NiP-2/Al2O3 catalyst, which can be attributed to the hydroxyl groups of AlO-H and PO-H species (Journal of Physical Chemistry 1990, 94, 5282-5285.), respectively, related to AlPO4 phase. These results strongly indicated the existence of AlPO4 phase. As the catalysts were reduced at 900 °C and DMR tests were performed at 700 °C, while the AlPO4 phase was usually reduced above 900 °C, the formed AlPO4 phase can not likely be reduced during DMR test. Therefore, AlPO4 phase also existed in the spent catalysts.

The description of the H2-TPR result is not accurate and revisions have been made in the manuscript (Line 289-290).

Point 11: Did you estimate quantitative acidity of catalysts by NH3-TPD? Which sample have the most acidity?

Response 11:

The acidity measurements of the catalysts were measured by NH3-TPD. Normally, the peak area of NH3-TPD curves is positively correlated with the content of catalysts acidity and the peak position is corresponding to the degree of acidity. After the calibration from impulsing with a given amount of NH3 gas to the detector of the TPD apparatus, a correction factor between peak area of NH3-TPD curves and the content of catalysts acidity can be obtained. The NH3-TPD analyses in this manuscript were performed by a commercial corporation, unfortunately, they didn’t provide the correction factor of NH3-TPD curves peak area to quantitate the catalysts acidity content. Thus we can only estimate the acidity content semiquantitatively by the NH3-TPD curves peak areas. The peaks area data are as follows:

Al2O3: 73.4, Ni/Al2O3: 50.6, NiP-4/Al2O3: 33.9, NiP-3/Al2O3: 29.1, NiP-2/Al2O3: 29.3.

According to the NH3-TPD profiles, pure Al2O3 has the most acidity. For the catalysts, Ni/Al2O3 exhibits the most acidity.

Point 12: To be honest, the difference in conversion values for various catalysts is within the margin of error.

Response 12:

It is really true as Reviewer suggested that the difference among the catalysts is insignificant. The initial methane and carbon dioxide conversions of Ni/Al2O3 catalyst are as high as 80% and 87%, which is close to the equilibrium conversion. A slight decrease of activity is observed for the P-containing catalysts in dry methane reforming. The CH4 conversion decreases to 76% and CO2 conversion decreases to the range of 83-85%, the variation values of the conversions are within 4%, which is indeed within the margin of error. It’s hard to say that there is an obvious difference among Ni/Al2O3 and NiP-x/Al2O3 catalyst in terms of conversions. However, the carbon deposition contents change distinctly after the addition of phosphorus. The content of carbon deposition decreases in the following order: Ni/Al2O3 (34.8%) > NiP-4/Al2O3 (21.8%) > NiP-3/Al2O3 (17.5%) > NiP-2/Al2O3 (2.7%), which strongly implies that the existence of P species results in better resistance of carbon deposition over the catalysts during the dry methane reforming process.

Point 13: You have two “Table 2” in your manuscript.

Response 13:

Thanks for the kind comment of the reviewer, the editing error has been revised in the manuscript.

Point 14: Pages 10-11. Table 2. A similar remark to remark 2.

Response 14:

Thanks for the kind comment of the reviewer, the particle size data should be round to tenths in Table 3 and the editing error has been revised in the manuscript.

Point 15: Lines 377-378. “The existence of AlPO4 phase was also confirmed by XPS spectra as considerable peaks attribute to P6+ determined by P 2p XPS spectrum” . Why P6+? What kind of XPS spectra are you talking about?

Response 15:

It is really true as the Reviewer suggested that the valence of P should be P5+ in phosphate species and Pδ- in nickel phosphide species. We have made corrections according to the Reviewer’s comments.

We tried our best to improve the manuscript and made some changes which will not influence the content and framework of the paper. We would like to take this opportunity to thank you for all your time involved and this great opportunity for us to improve the manuscript. We hope you will find this revised version satisfactory.

Sincerely,

Tong LI, Hongfeng YIN

Reviewer 2 Report

This study is interesting and the way the author carried out addressing each point is really needed to be appreciated.  The work design and characterization techniques present the effect give justice to the work done and it can be published. However, there are other reports on the same NiP-x/Al2O3 catalysts. Compare the reported results with this current study. In addition, here are some minor points suggest for the author's consideration.

  1. The introduction part is very short. Comparison with other catalysts like SSZ-13 zeolite, Ni-ZrO2@SiO2, MgO-Ni/Al2O3 is highly suggested.
  2. How does the P loading resist the carbon deposition to improve the anti-coking property? Explain the mechanism.
  3. There are many English grammatical and typo errors. Please correct it carefully. Example. “The impregnated sample were dried at 110 °C over night,”.

Author Response

Dear Reviewer,

Thanks for your detailed review concerning our manuscript. These comments are valuable for revising and improving our paper with important guiding significance. We have made corrections according to the comments, revised portions are marked up using the “Track Changes” function of MS Word.

 “However, there are other reports on the same NiP-x/Al2O3 catalysts. Compare the reported results with this current study.”

Thanks for your great suggestion on improving the accessibility of our manuscript. The relevant contents are provided below for your quick reference:

Line 111 to Line 115: González-Castaño et al.37 investigated the catalytic performance of 20 wt% Ni2P supported on Al2O3, CeO2 and SiO2-Al2O3, the result indicated that the Ni2P/Al2O3 catalyst showed highest conversion and stability. However, visible deactivation was still observed after 10 h of time on stream, which may be due to the oxidation of Ni2P phase during dry methane reforming.

To facilitate this discussion, we first retype your comments in italic font and then present our responses to the comments in red.

Point 1: The introduction part is very short. Comparison with other catalysts like SSZ-13 zeolite, Ni-ZrO2@SiO2, MgO-Ni/Al2O3 is highly suggested.

Response 1:

Thanks for your great suggestion on improving the accessibility of our manuscript., the comparison with catalysts in dry methane reforming has been further described in the introduction part. The relevant contents are provided below for your quick reference:

1) The carbon deposition formation mechanism was described at Line 36 to Line 45:

As is shown in the literatures7, carbon deposition on the surface of nickel catalyst is generally carried out according to the following steps: methane molecules dissociate on the metal surface to generate adsorbed CHx-* species, which will be further dehydrogenated to form C-*; part of the C-* species can be removed by the surface active O-* species generated from the dissociation of CO2 molecules and converted into CO; at the same time, the unconverted C-* will further undergo deep cracking on the metal surface to form surface carbon. In addition, the further aggregation of adsorbed CHx-* species is another way to form carbon deposits. It follows that increasing the adsorption and dissociation rate of CO2 and appropriately decreasing the CH4 dissociation ability of the catalyst are effective ways to inhibit the carbon deposition on the surface of nickel catalyst.

2) The comparison with catalysts in dry methane reforming has been further described at Line 47 to Line 75:

These studies mainly focus on: 1) Enhance the surface basicity of the catalysts by the addition of alkaline promoters, which increases the CO2 dissociation ability, promotes the generation of surface-active O-* species and enhances the elimination ability of surface C-* species on nickel catalysts. The addition of alkalis8, alkaline-earth metals9, 10 and rare-earth metals (La, Ce)11, 12, can enhance the dispersion of nickel species and the surface basicity and promote the generation of surface active O-* species from CO2, which results in better carbon deposition resistance over nickel catalysts. Whereas, the enhancement of surface basicity always accompanied with the decrease of methane dissociation rate, which leads to the decrease of DMR activity8; 2) Obtain smaller nickel nanoparticle size by confining nickel species with mesoporous materials13, 14, inorganic shells11, 15, and the edge of the support3, which is beneficial of the resistance to carbon formation during DMR. Wang et al.13 investigated S-2 zeolite fixed nickel nanoparticles structure catalyst (Ni@S-2) in DMR and found Ni@S-2 exhibited superb carbon resistance. Liu et al.16 synthesized a multiple-core@shell structured catalyst (Ni-ZrO2@SiO2) catalyst and investigated its catalytic performance in DMR. Due to the small nickel nanoparticle size and the confinement effect of SiO2 shell, Ni-ZrO2@SiO2 displayed ultra-high coking resistance for dry methane reforming. Song et al. found that the edges of MgO crystals exhibited good properties of the stabilization of nickel nanoparticles, which resulted in good resistance of carbon deposition in DMR. What’s more, special structure catalysts, such as spinel(NiAl2O417, Ni/MgAl2O418), perovskite (LaNiO319, La1−xPrxNiO3−δ20, La0.6Sr0.2Ti0.85Ni0.15O3-δ21), hydrotalcites (Ce-NiMgAl hydrotalcite22, Zr-NiMgAl hydrotalcite23, La-NiMgAl hydrotalcite24) and solid solution (NiOMgO25, NiO-CeO226), which had strong ability of confining nickel nanoparticles, are also used to increase the carbon resistance of nickel catalysts; 3) Decrease the outer electron cloud density of nickel catalyst by alloying nickel with other metals27, 28, which decreases CH4 dissociation ability of the catalysts. Turap et al.27 indicated that Co–Ni alloy promoted the adsorption of surface oxygen and enhanced carbon removal, resulting in less carbon deposition compared to nickel catalyst.; 4) Poison CH4 dissociation ability of nickel with sulfur29, which, however, passivates the activity of nickel catalysts in dry methane reforming. 

Point 2: How does the P loading resist the carbon deposition to improve the anti-coking property? Explain the mechanism.

Response 2:

Carbon deposition on the surface of nickel catalyst is generally carried out according to the following steps: methane molecules dissociate on the metal surface to generate adsorbed CHx-* species, which will be further dehydrogenated to form C-*; part of the C-* species can be removed by the surface active O-* species generated from the dissociation of CO2 molecules and converted into CO; at the same time, the unconverted C-* will further undergo deep cracking on the metal surface to form surface carbon. In addition, the further aggregation of adsorbed CHx-* species is another way to form carbon deposits. It follows that increasing the adsorption and dissociation rate of CO2 and appropriately reducing the methane dissociation ability of the catalyst are effective ways to inhibit the carbon deposition on the surface of nickel catalyst.

Phosphorus loading on the nickel catalyst results in the formation of nickel phosphides species, which increases the CO2 dissociation ability of the catalyst. In-situ CO2-DRIFTS analysis indicates the enhancement of CO2 dissociation ability of NiP-2/Al2O3 catalyst. Compared to Ni/Al2O3, the spectrum of NiP-2/Al2O3 catalyst presents evident higher intensity of the bands belonging to adsorbed CO2 species, which implies the higher CO2 adsorption ability of NiP-2/Al2O3 catalyst. Additionally, lower bicarbonates bands at 1652 and 1430 cm1 and higher formates species bands at 1562, 2931 and 2862 cm1 over NiP-2/Al2O3 catalyst are also observed. This result indicates that phosphorus presence catalysts own better activation ability of CO2. The produced bicarbonates species are more easily further converted to formate intermediates and active O* species, which favors the elimination of the carbon deposition.

Point 3: There are many English grammatical and typo errors. Please correct it carefully. Example. “The impregnated sample were dried at 110 °C over night,”.

Response 3:

We are very sorry for our grammatical and typo errors, and we have checked the manuscript and corrected it carefully.

We would like to take this opportunity to thank you for all your time involved and this great opportunity for us to improve the manuscript. We hope you will find this revised version satisfactory.

Sincerely,

Tong LI, Hongfeng YIN

Round 2

Reviewer 1 Report

Dear authors, thank you for improving paper quality.